# Towards High-performance Spiking Transformers from ANN to SNN Conversion

## ABSTRACT

Spiking neural networks (SNNs) show great potential due to their energy efficiency, fast processing capabilities, and robustness. There are two main approaches to constructing SNNs. Direct training methods require much memory, while conversion methods offer a simpler and more efficient option. However, current conversion methods mainly focus on converting convolutional neural networks (CNNs) to SNNs. Converting Transformers to SNN is challenging because of the presence of non-linear modules. In this paper, we propose an Expectation Compensation Module to preserve the accuracy of the conversion. The core idea is to use information from the previous T time-steps to calculate the expected output at time-step T. We also propose a Multi-Threshold Neuron and the corresponding Parallel Parameter normalization to address the challenge of large time steps needed for high accuracy, aiming to reduce network latency and power consumption. Our experimental results demonstrate that our approach achieves state-of-the-art performance. For example, we achieve a top-1 accuracy of 88.60% with only a 1% loss in accuracy using 4 time steps while consuming only 35% of the original power of the Transformer. To our knowledge, this is the first successful ANN to SNN conversion for Spiking Transformers that achieves high accuracy, low latency, and low power consumption on complex datasets.

## CCS CONCEPTS

• **Computing methodologies → Artificial intelligence**.

## KEYWORDS

Spiking Neural Networks, Spiking Transformer, ANN-SNN Conversion, Expectation Compensation, Multi-Threshold Neurons

## 1 INTRODUCTION

Spiking neural networks(SNNs) are a type of neural network model that imitates the mechanisms of biological neurons[1, 17]. They are called the third generation of neural networks [32] due to their biologically plausible interpretation and efficient computational efficiency[41, 49]. Unlike traditional neural networks, SNNs concentrate on the generation and reception of spikes. Neurons in SNNs do not produce output in every iteration. Instead, they become active and emit spikes only when their membrane potential

*ACM MM, 2024, Melbourne, Australia*

© 2024 Copyright held by the owner/author(s). Publication rights licensed to ACM.
ACM ISBN 978-x-xxxx-xxxx-x/YY/MM
https://doi.org/10.1145/nnnnnnn.nnnnnnn

reaches a specific threshold. This sparse spike activity results in significantly higher computational efficiency than traditional neural networks [37], especially when deployed on neuromorphic chips [6, 7, 33]. However, training large-scale, high-precision, and low-latency SNNs remains challenging due to the non-differentiable nature of spikes.

Currently, there are two main approaches to train SNNs. The first approach is direct training using backpropagation [14, 26, 34, 46, 47, 53]. This method employs surrogate gradients during backpropagation, which utilizes differentiable continuous functions or spike-time-dependent plasticity strategies to replace the non-differentiable spike emission rules. However, this training process still relies on standard GPUs that are not well-suited for the unique characteristics of SNNs, leading to significant resource consumption and limited performance. The second approach is Artificial Neural Network (ANN) to SNN conversion [3, 4, 9, 28, 38]. This conversion method does not require any additional training. Instead, it uses pre-trained ANNs and replaces the activation functions with spiking neurons. This process takes advantage of the similarity between ReLU activation functions and the spike emission rates of integrate-and-fire models. The result is a conversion of ANNs into SNNs while significantly preserving the original ANN's performance. However, this method often leads to longer inference times, and the modules that can be successfully converted are limited.

As is well known, Transformers have demonstrated exceptional performance in various vision tasks [5, 12, 23, 31, 35]. However, despite numerous efforts to convert CNNs to SNNs, no well-established method exists for converting Transformer models. This is because Transformers have modules such as layernorm and GELU that differ from the ReLU function in CNNs. These modules require interaction between neurons within the same layer and exhibit non-linear characteristics, making it challenging to achieve accurate conversion through the linear piecewise quantization of individual neurons.

This paper proposes a new method to convert Transformer to SNN. The primary obstacle in this conversion is dealing with non-linear modules. To overcome this challenge, we propose using an Expectation Compensation Module (ECM) that calculates expectations and replaces each non-linear module. Specifically, a customized ECM is employed in place of the matrix product, conducting most of its operations through accumulations. This reduces power consumption and ensures that the total output matches the expected result at each moment. To improve the efficiency of minimal spikes, we introduce Multi-Threshold Neurons and the corresponding Parallel Parameter normalization, significantly reducing the required latency and power consumption for inference with comparable accuracy.

Our main contributions are summarized as follows:

- We analyze the challenges of non-linear module conversion in Transformer and present a novel solution called the Expectation Compensation Module, which uses the information from the previous time steps to calculate the expected

output at the current time step, thus effectively tackling the limitations of conventional conversion methods while demonstrating a minimal power consumption increase.

- To overcome the issue of slow accuracy improvement over time during Transformer conversion, we propose a Multi-Threshold Neuron and the corresponding Parallel Parameter normalization, substantially reducing power consumption requirements and significantly decreasing latency.
- The proposed method has shown its effectiveness on the ImageNet1k dataset. Our approach not only outperforms the accuracy of existing SNN models but also substantially reduces power consumption compared to other Transformer models. For example, we reach a top-1 accuracy of 88.60%, with a minimal accuracy loss of only 1% compared to ANN , while achieving a 65% reduction in energy consumption.

## 2 RELATED WORKS

### ANN-SNN Conversion

The ANN-SNN conversion methods aim to replicate the performance of ANNs by converting pre-trained ANN weights into synaptic weights of SNNs. Initially, Cao et al. [4] proposed to train an ANN with ReLU activation function and then use IF neurons to replace the activation layer in the conversion process. Diehl et al. [10] further narrowed the gap between ANNs and SNNs by scaling and normalizing the weights. To address spike count errors resulting from the hard reset mechanism, a soft reset neuron was proposed in Rueckauer et al. [38] and Han et al. [18] to preserve temporal information.

Further research has conducted both theoretical and experimental examinations of conversion errors and investigated different optimization techniques to minimize these errors, including: (1) Optimizing thresholds: Sengupta et al. [39] and Zhang et al. [50] proposed dynamic threshold adjustment strategies during the conversion process. (2) Optimizing membrane potential: Bu et al. [2] demonstrated that setting the initial membrane potential at half the threshold can reduce conversion errors. They also proposed an analysis of residual membrane potential as a strategy to eliminate conversion errors [20]. (3) Optimizing the pre-conversion ANN structure: Esser et al. [13] suggested training ANNs with quantized activation values. Ho and Chang [21] introduced a trainable clipping layer (TCL) for threshold determination. Ding et al. [11] proposed a rate norm layer as a replacement for ReLU in ANN training, while [3, 19, 24, 43] introduced different activation functions to replace ReLU in ANN training. (4) Optimizing spiking neuronal models. Li et al. [30] introduced a neuron model capable of releasing burst spikes. Wang et al. [44] proposed a memory-enhanced signed neuron model, while Li et al. [27] suggested incorporating negative spikes and extending simulation time, which indicates that increasing the precision of the output layer can improve accuracy at a relatively small cost.

The Previous approaches for converting CNNS to SNNS were restricted by the CNNS' performance. Jiang et al.[25] introduced Universal Group Operators and a Temporal-Corrective Self-Attention Layer to approximate the original Transformer. However, it has a long inference latency and a gap with the ANN.

In contrast, this paper presents a new method for converting a Transformer to an SNN and demonstrates that this conversion method can achieve high accuracy and low latency while reducing network energy consumption.

## Directly Trained Transformer in ANNs and SNNs

The Transformer architecture has performed exceptionally in the ANN and SNN domains. Initially, Transformers gained prominence in the ANN domain due to their outstanding performance with self-attention mechanisms. The original Transformer architecture, which consists of an encoder and a decoder, was proposed by Vaswani et al.[42]. Upon this, Dosovitskiy et al. [12] introduced the Vision Transformer (ViT) model in 2020. The ViT model divides images into fixed-size patches as token inputs, and this successful application of the Transformer architecture in computer vision has resulted in significant achievements. Fang et al. [15, 16] and Sun et al. [40] have further scaled standard ViT models to one billion parameters, exploring the performance limits of large-scale visual models.

In the SNN domain, research on spike-based Transformers quickly emerged. Researchers have proposed different SNN-based spike self-attention mechanisms, albeit with some floating-point calculations [29, 55]. Subsequently, Zhou et al. [54] and Yao et al. [48] introduced fully event-driven Transformers, while Wang et al. [45] enhanced the computational efficiency and accuracy of Spiking Transformers by introducing masking techniques. Previous studies have successfully applied the combination of Transformers and SNNs to a range of applications, such as monocular depth estimation [52], single-object tracking with event cameras [51], and automatic speech recognition. Wang et al.[45] first trained a modified Transformer and then converted it into a Spiking Transformer.

In contrast to the methods mentioned above that train Transformer networks from scratch, this paper focuses on converting pre-trained Transformer networks into SNNs to reduce energy consumption while preserving the model's performance.

## 3 PRELIMINARIES

In this section, we first detail the theoretical basis of the conversion process from ANNs to SNNs. Then, we introduce the Vision Transformer (ViT), the ANN architecture we selected for conversion.

### 3.1 ANN-SNN conversion theory

*Neurons in ANNs.* In ANNs, for linear or convolution layers in CNNs that use the ReLU activation, the output $a^l$ of neurons in layer $l$ can be formulated as:

$$a^l = \text{ReLU}(W^l a^{l-1}) = \max(W^l a^{l-1}, 0), \quad (1)$$

where $W^l$ denotes the linear transformation or convolution weights in this layer.

*Integrate-and-Fire Neurons in SNNs.* For Integrate-and-Fire (IF) neurons in SNNs, let $m^l(t)$ and $v^l(t)$ denote the membrane potential of neurons in the $l$-th layer before and after firing spikes at time-step

$t$, the neural dynamic can be formulated as follows:

$$\boldsymbol{m}^l(t) = \boldsymbol{v}^l(t-1) + \boldsymbol{W}^l \boldsymbol{x}^{l-1}(t), \tag{2}$$

$$\boldsymbol{s}^l(t) = H(\boldsymbol{m}^l(t) - \theta^l), \tag{3}$$

$$\boldsymbol{x}^l(t) = \theta^l \boldsymbol{s}^l(t), \tag{4}$$

$$\boldsymbol{v}^l(t) = \boldsymbol{m}^l(t) - \boldsymbol{x}^l(t). \tag{5}$$

where $H$ represents the Heaviside step function and $\theta^l$ denotes the threshold of the neuron in layer $l$. $\boldsymbol{s}^l(t)$ is the actual output spike of layer $l$. $\boldsymbol{x}^l(t)$ is the postsynaptic potential and theoretical output of layer $l$, the element of which equals $\theta^l$ if the neuron fires and 0 otherwise. Similar to [38] and [18], we use the "reset-by-subtraction" mechanism, where the membrane potential $\boldsymbol{v}^l(t)$ decreases by a value of $\theta^l$ if the neuron fires.

*ANN-SNN Conversion.* Combining Equations (2)-(5), we have

$$\boldsymbol{v}^l(t) - \boldsymbol{v}^l(t-1) = \boldsymbol{W}^l \boldsymbol{x}^{l-1}(t) - \boldsymbol{x}^l(t). \tag{6}$$

By summing from time-step 1 to time-step $T$, we have

$$\frac{\boldsymbol{v}^l(T) - \boldsymbol{v}^l(0)}{T} = \frac{\boldsymbol{W}^l \sum_{i=1}^{T} \boldsymbol{x}^{l-1}(i)}{T} - \frac{\sum_{i=1}^{T} \boldsymbol{x}^l(i)}{T}. \tag{7}$$

Let $\Phi^l(T) = \frac{\sum_{i=1}^{T} \boldsymbol{x}^l(i)}{T}$, then we have

$$\Phi^l(T) = \boldsymbol{W}^l \Phi^{l-1}(T) - \frac{\boldsymbol{v}^l(T) - \boldsymbol{v}^l(0)}{T}. \tag{8}$$

By comparing Equation (1) and Equation (8), it can be observed that the term $\frac{\boldsymbol{v}^l(T) - \boldsymbol{v}^l(0)}{T}$ tends to 0 when $T$ is sufficiently large. This allows us to use $\Phi^l(T)$ in SNNs to approximate $\boldsymbol{a}^l$ in ANNs.

*Parameter normalization.* Approximation errors are inevitable due to the nature of spike-based communication between neurons since the neurons in SNNs can emit one spike at each time and are therefore limited to a firing rate range of $[0, r_{\max}]$, whereas ANNs typically do not have such constraints. To prevent approximation errors caused by excessively low or high firing rates, weight normalization was introduced by [10, 38]. This normalization approach rescales all parameters using the following equations:

$$W_{\text{SNN}}^l = W_{\text{ANN}}^l \frac{\lambda^{l-1}}{\lambda^l}. \tag{9}$$

The value of $\lambda^l$ is determined by the $p$-th percentile of the total activity distribution of layer $l$. Modifying Equation (9) and setting $\theta_j^l$ to 1 is equivalent to adjusting the firing threshold on the soft-reset neuron to $\lambda^l$[2]. This adjustment ensures that the output $\boldsymbol{x}^l(t)$ is a spike matrix equal to $\boldsymbol{s}^l(t)$ and suits the operational dynamics of SNNs.

## 3.2 Vision Transformer

Vision Transformer (ViT) architecture consists of three core components: Embeddings, Transformer Encoder, and Classification Head.
**Embeddings**: The process starts by segmenting an image into patches of specific dimensions, viewing them as a sequence of tokens. Each patch undergoes linear embedding with added positional embeddings, enriching the output token vectors with the patch's content and location within the image.

**Transformer Encoder**: Central to feature extraction, the Transformer Encoder plays a crucial role in various visual tasks. It is divided into two primary segments:

(1) Self-Attention Mechanism. This mechanism calculates a weighted sum of all the values $V$ in a given sequence. The attention weights are determined based on the similarity between a query $Q$ and a key $K$. The values $Q$, $K$, and $V$ are obtained through the input $X$ using weight matrices $W^Q$, $W^K$, and $W^V$ respectively. The following equation describes the matrix form of the output calculation for the self-attention mechanism:

$$O = \text{Softmax}\left(\frac{Q^T K}{\sqrt{d}} V\right) = \text{Softmax}\left(\frac{(W^Q X)^T W^K X}{\sqrt{d}} W^V X\right). \tag{10}$$

where $d$ is the dimension of the key and query vectors.

(2) Feed-Forward Network. Here, the input vector passes through two linear layers and is activated by the GELU function between them.
**Classification Head**: Features related to the CLS token are directed toward the classification head, which then computes the probabilities for the various classes.

## 4 METHOD

In this section, we first analyze the main errors encountered in ANN-SNN conversion. Following this, we propose the Expectation Compensation Module (EC) to preserve the accuracy of non-linear modules. In particular, we detailed a lossless conversion method for the matrix product layer, mainly using additional operations. Additionally, a Multi-Threshold Neuron (MT) is designed to improve the efficiency of minimal spikes, which significantly reduces network latency and energy consumption. The diagram shown in Figure 1 provides an overview of the architecture we utilized.

## 4.1 Error Analysis of Nonlinear Module in ANN-SNN Conversion

Existing ANN-SNN conversion methods mainly focus on CNNs, which typically employ linear operations, such as linear transformations and convolutions, combined with ReLU activation, as formulated in Equation (1). However, Transformer architecture uses many non-linear operations, such as GELU, softmax, layernorm, and matrix product, which cannot be directly formulated using Equation (1). Consequently, the current conversion theory discussed in Section 3.1 does not apply to Transformers, which can lead to conversion errors.

To be specific, we assume that the outputs of layer $l-1$ in both ANNs and SNNs are identical, denoted as $\boldsymbol{a}^{l-1} = \Phi^{l-1}(T) = \frac{\sum_{t=1}^{T} \boldsymbol{x}^{l-1}(t)}{T}$, and we will compare the outputs $\boldsymbol{a}^l$ and $\Phi^l$ in layer $l$.

Considering an arbitrary non-linear module in layer $l$ of an ANN, its function can be formulated as:

$$\boldsymbol{a}^l = F(\boldsymbol{a}^{l-1}), \tag{11}$$

where $F$ is the function of this layer. Obviously, it cannot be expressed equivalently using Equation (1). In this case, if we do not introduce a further conversion method for this non-linear module, the actual output of the SNN counterpart at time $t$ will be

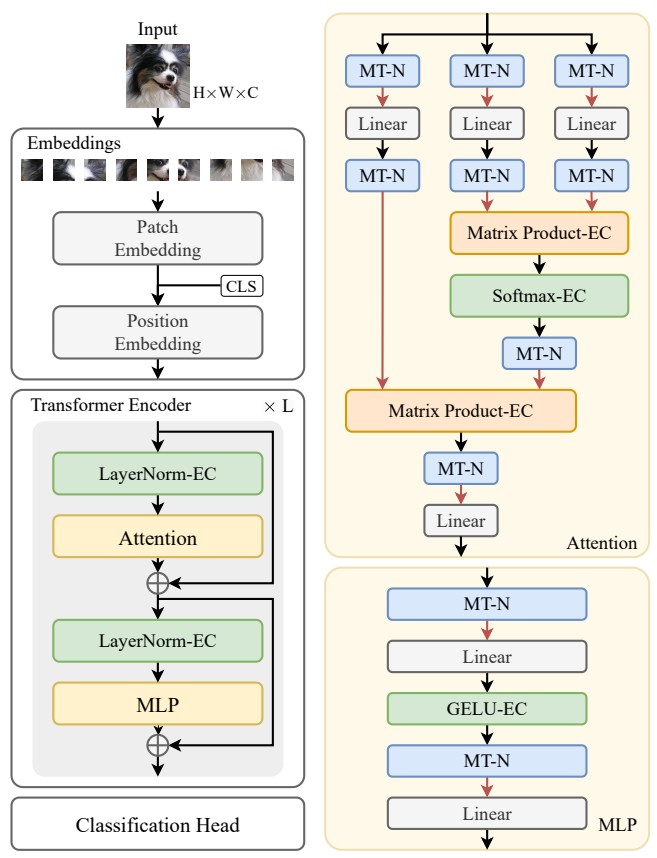

Figure 1: An overview of the proposed architecture, including the whole architecture, Attention, and MLP module.

$x^l(t) = F(x^{l-1}(t))$. The average output can be formulated as follows:

$$\Phi^l(T) = \frac{\sum_{t=1}^{T} x^l(t)}{T} = \frac{\sum_{t=1}^{T} F(x^{l-1}(t))}{T}. \qquad (12)$$

However, in the case of ANNs, the expected average output can be formulated as:

$$a^l = F(a^{l-1}) = F\left(\frac{\sum_{t=1}^{T} x^{l-1}(t)}{T}\right). \qquad (13)$$

Due to the non-linear nature of the module, we have:

$$\frac{\sum_{t=1}^{T} F(x^{l-1}(t))}{T} \neq F\left(\frac{\sum_{t=1}^{T} x^{l-1}(t)}{T}\right). \qquad (14)$$

This implies that the output $\Phi^l(T)$ of SNNs in Equation (12) is not equivalent to the output $a^l$ of ANNs in Equation (13), posing challenges for non-linear conversion.

## 4.2 Expectation Compensation Module

To overcome the challenge of converting non-linear layers, we propose using Expectation Compensation Modules to preserve non-linearity throughout the conversion process by leveraging prior information to compute expectations.

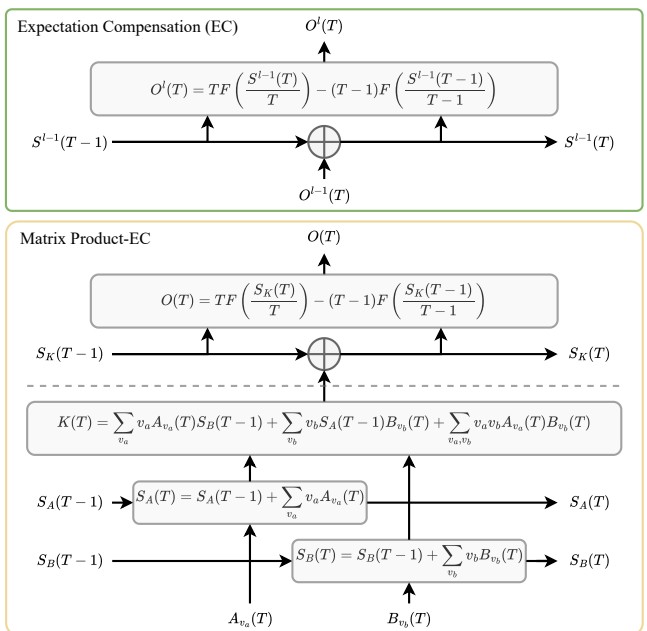

Figure 2: The upper diagram shows the general Expectation Compensation module(EC). The lower diagram shows the Expectation Compensation module for Matrix Product(Matrix Product-EC).

### 4.2.1 General Expectation Compensation Module.

The theorem below calculates the expected output of the arbitrary non-linear layer at each time step in SNNs.

THEOREM 4.1. *Consider a non-linear layer $l$ with a function $F$. In SNNs, the output of this layer at time $t$ is denoted as $O^l(t)$. Let $S^l(T)$ be the cumulative sum of layer $l$ outputs up to time $T$, given by $S^l(T) = \sum_{t=1}^{T} O^l(t)$. The expected output of the SNNs at time $T$ is given by:*

$$O^l(T) = TF\left(\frac{S^{l-1}(T)}{T}\right) - (T-1)F\left(\frac{S^{l-1}(T-1)}{T-1}\right). \qquad (15)$$

The detailed proof is provided in the supplementary materials. Theorem 4.1 indicates that lossless conversion can be achieved by an accumulator to records $S^{l-1}(T)$ and an optional variable to records $TF\left(S^{l-1}(T)/T\right)$ as shown in Figure 2.

### 4.2.2 Expectation Compensation Module for Matrix Product.

For the matrix product layer, we can convert it into a specialized module that primarily uses additional operations to achieve lossless conversion. The theorem below outlines how to calculate the expected output of the matrix product layer at each time step in SNNs.

THEOREM 4.2. *Consider a module for matrix product that receives two sets of spike inputs, denoted by $A_{v_a}(t)$ and $B_{v_b}(t)$. These inputs are generated by neurons A and B, respectively, and are characterized by multiple thresholds $v_a$ and $v_b$, as described in Section 4.3.*

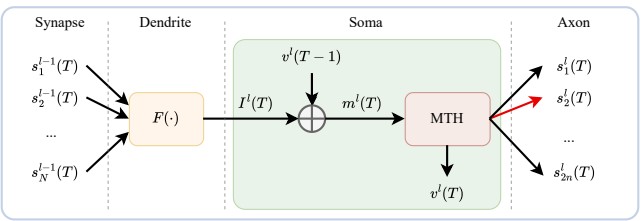

Figure 3: Diagram of MT neuron. MT neuron receives input from nonlinear/linear modules and emits up to one spike.

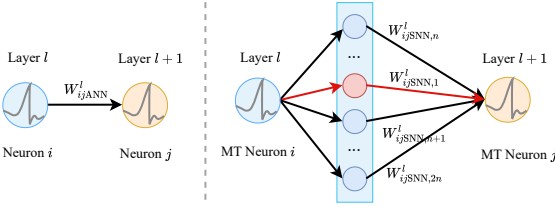

Figure 4: Left: Original connection in ANN. Right: Parallel Parameter normalization of MT neuron in SNN. The MT Neuron extends one connection to $2n$ channels. At each time, only one of the $2n$ channels can emit a spike.

We can integrate the input by $A(t) = \sum_{v_a} v_a A_{v_a}(t)$ and $B(t) = \sum_{v_b} v_b B_{v_b}(t)$. Here, $A(t)$ and $B(t)$ are the sum matrices weighted by multiple thresholds $v_a$ and $v_b$, respectively.

Let $S_A(T) = \sum_{t=1}^{T} A(t)$ and $S_B(T) = \sum_{t=1}^{T} B(t)$ represent the cumulative sum of inputs up to time $T$. We define $S_K(T) = S_A(T)S_B(T)$. Then, the expected output at time $T$ can be formulated as:

$$O(T) = \frac{1}{T}S_K(T) - \frac{1}{T-1}S_K(T-1), \tag{16}$$

where $S_K(T)$ can be calculated mainly using addition, as described by the following equation:

$$S_K(T) = S_K(T-1) + K(T) \tag{17}$$

$$K(T) = \sum_{v_a,v_b} v_a v_b A_{v_a}(T) B_{v_b}(T) + \sum_{v_a} v_a A_{v_a}(T) S_B(T-1)$$
$$+ \sum_{v_b} v_b S_A(T-1) B_{v_b}(T). \tag{18}$$

The detailed proof is provided in the supplementary materials. According to Theorem 4.2, the output $O(T)$ can be obtained through the process illustrated in Figure 2. The main power consumption in this process occurs during the matrix product calculation of $K(T)$ using spike matrices, which can be implemented through accumulations. Since each position of the input matrix has only one effective threshold at each time, it limits the total number of input spikes, thereby restricting the total number of operations. Combined with the sparsity of spikes, this reduces power consumption at each time step while achieving lossless conversion.

### 4.3 Multi-Threshold Neuron

#### 4.3.1 Problem of Consumption and Latency.

If we only use the Expectation Compensation Module, neuron communication will remain in a floating-point format. As discussed in Section 5.5, most of the network's power consumption occurs in the linear and matrix product layers. To reduce the network's energy consumption, we introduce spiking neurons before each linear layer and matrix product layer. Thus, we can significantly reduce the network's power consumption by adopting spiking communication.

However, if we only use one threshold, no matter how set, it will result in excessively high firing rates or high inference latency. The findings in Section 5.4 demonstrate the importance of having large and small thresholds in the Transformer.

#### 4.3.2 The Proposed Solution: Multi-Threshold Neuron.

To tackle the challenges of high power consumption and latency, we propose a Multi-Threshold Neuron (MT neuron).

This neuron model has additional thresholds built upon the base threshold, allowing it to process more information in a single time step. The MT neuron is characterized by parameters including the positive and negative base thresholds, represented as $\theta_1$ and $-\theta_2$, respectively, and the number of thresholds denoted as $2n$. We can refer to $\lambda_p^l$ as the $p$-th threshold value of the MT neuron corresponding to index $p$.

$$\lambda_1^l = \theta_1^l, \lambda_2^l = 2\theta_1^l, ..., \theta_n^l = 2^{n-1}\theta_1^l,$$
$$\lambda_{n+1}^l = -\theta_2^l, \lambda_{n+2}^l = -2\theta_2^l, ..., \lambda_{2n}^l = -2^{n-1}\theta_2^l, \tag{19}$$

As shown in Figure 3, the dynamic of MT neurons is described by:

$$I_j^l(t) = F_j^l(s_{,1}^{l-1}(t), ..., s_{,2n}^{l-1}(t)), \tag{20}$$

$$m_j^l(t) = v_j^l(t-1) + I_j^l(t), \tag{21}$$

$$s_{j,p}^l(t) = MTH_{\theta_1,\theta_2,n}(m_j^l(t)) \tag{22}$$

$$x_j^l(t) = \sum_p s_{j,p}^l(t)\lambda_p^l, \tag{23}$$

$$v_j^l(t) = m_j^l(t) - x_j^l(t). \tag{24}$$

The variables $I_j^l(t), s_j^l(t), x_j^l(t), m_j^l(t)$ and $v_j^l(t)$ respectively represent the input, output, postsynaptic potential, and the membrane potential before and after spikes of the $j$-th neuron in the $l$-th layer at time $t$. Meanwhile, $F$ is a linear or nonlinear function of this layer. The function $MTH_{\theta_1,\theta_2,n}(x)$ can be described using the following piecewise function:

$$MTH_{\theta_1,\theta_2,n}(x):$$

$$\begin{cases} \lambda_n^l - \frac{\lambda_1^l}{2} \leq x: & s_{j,n}^l(t) = 1, \\ \lambda_{n-1}^l - \frac{\lambda_1^l}{2} \leq x < \lambda_n^l - \frac{\lambda_1^l}{2}: & s_{j,n-1}^l(t) = 1, \\ ... & ... \\ \frac{\lambda_1^l}{2} \leq x < \lambda_2^l - \frac{\lambda_1^l}{2}: & s_{j,1}^l(t) = 1, \\ \frac{\lambda_{n+1}^l}{2} \leq x < \frac{\lambda_1^l}{2}: & \text{all the } s_{j,p}^l(t) = 0, \\ \lambda_{n+2}^l - \frac{\lambda_{n+1}^l}{2} \leq x < \frac{\lambda_{n+1}^l}{2}: & s_{j,n+1}^l(t) = 1, \\ ... & ... \\ \lambda_{2n}^l - \frac{\lambda_{n+1}^l}{2} \leq x < \lambda_{2n-1}^l - \frac{\lambda_{n+1}^l}{2}: & s_{j,2n-1}^l(t) = 1, \\ x < \lambda_{2n}^l - \frac{\lambda_{n+1}^l}{2}: & s_{j,2n}^l(t) = 1. \end{cases} \tag{25}$$

The results of experiments presented in Section 5.4 indicate that although this neuron has multiple thresholds, most of the spikes it generated are concentrated in $\theta_1$ and $-\theta_2$. The spikes generated by other thresholds are minimal, which reduces energy consumption and inference latency.

#### 4.3.3 Parallel Parameter normalization for MT Neuron.

Spike neurons communicate with each other by producing an output spike of either 0 or 1. As for function $F$ in Figure 3.

If $F$ is a Matrix Product-EC function, we only need to send spikes $s^l(t)$ to $F$ as $\boldsymbol{A}_{v_a}(t)$ or $\boldsymbol{B}_{v_b}(t)$.

If $F$ is a general nonlinear EC function, we will integrate spike output by $I_j^l(t) = F_j^l(\sum_p \boldsymbol{s}_{,p}^{l-1}(t)\lambda_p^{l-1})$.

If $F$ is a linear function, $I_j^l(t)$ can be expressed by

$$I_j^l(t) = \sum_i w_{ij\text{ANN}}^l x_i^{l-1}(t) = \sum_i w_{ij\text{ANN}}^l \sum_p s_{i,p}^{l-1}(t)\lambda_p^{l-1} \quad (26)$$

A parallel parameter normalization method is proposed to support spike communication between MT neurons in a linear layer. This method extends the ANN weight to 2n weights in the SNN corresponding to 2n thresholds of MT neurons, as shown in Figure 4. We update these weights using the following formula:

$$W_{\text{SNN},p}^l = W_{\text{ANN}}^l \frac{\lambda_p^{l-1}}{\lambda_1^l} \quad (27)$$

Here, we divide an extra variable $\lambda_1^l$ to equilibrate parameter size. Let's set $\eta^l = \frac{\theta_2^l}{\theta_1^l}$. This brings the neuron to an equivalent form, which is as follows:

$$I_j^l(t) = \sum_{i,p} w_{ij\text{SNN},p}^l s_{i,p}^{l-1}(t) \quad (28)$$

$$\theta_{1,new} = 1, \theta_{2,new} = \eta \quad (29)$$

Based on the above discussion, we name this method: Expectation Compensation and Multi-Threshold(ECMT). The overall conversion algorithm can be summarized in Algorithm 1.

## 5 EXPERIMENTAL RESULTS

In this section, we first evaluate the proposed method's performance on the ImageNet dataset. Then, we compare our method with state-of-the-art SNN training and ANN-SNN conversion methods. Additionally, we perform ablation experiments on Multi-Threshold Neurons. Finally, we analyze the power consumption of the SNNs converted by our method.

### 5.1 Experimental Setup

We convert pre-trained Vision Transformer including the ViT-S/16, ViT-B/16, ViT-L/16 with 224 resolution [42], and the EVA model [16] on Imagenet1k dataset [8]. For all Multi-Threshold Neurons, we set $n$ to 8 for ViT-S/16, ViT-B/16, ViT-L/16 and 6 for EVA. And we set threshold percent $p$ to 99. A more detailed setup can be found in supplementary materials.

### 5.2 Experimental results on different model

Based on the provided data, Table 1 compares performance metrics for various architectures. The analysis shows that our SNN

---

**Algorithm 1** The conversion method using Expectation Compensation Module and Multi-Threshold Neuron(ECMT)

---

**Input**: Pre-trained Transformer ANN model $f_{\text{ANN}}(\boldsymbol{W})$; Dataset D; Time-step T to test dataset; Threshold percent p.

**Output**: SNN model $f_{\text{SNN}}(\boldsymbol{W}, \boldsymbol{\theta}_1, \boldsymbol{\theta}_2, \boldsymbol{v})$

1: **step1:** Obtain the base thresholds $\boldsymbol{\theta}_1$ and $\boldsymbol{\theta}_2$
2: **for** length of Dataset $\boldsymbol{D}$ **do**
3:     Sample minibatch data from $\boldsymbol{D}$
4:     Run the data on $f_{\text{ANN}}$ and static the activation values before linear and matrix product module at p% and (1-p%), setting them as $\boldsymbol{\theta}_1$ and $-\boldsymbol{\theta}_2$ respectively.
5: **end for**
6: **step2:** Converted to SNN model
7: **for** module m in $f_{\text{ANN}}$.Module **do**
8:     **if** m is Linear Module **then**
9:         Add a Multi-Threshold Neuron before m
10:     **else if** m is Matrix Product **then**
11:         replace m by two Multi-Threshold Neurons followed by a Matrix Product EC Module
12:     **else if** m is Other Nonlinear Module **then**
13:         replace m by an EC Module
14:     **end if**
15: **end for**
16: Set the base thresholds of MT neurons to corresponding $\boldsymbol{\theta}_1, -\boldsymbol{\theta}_2$ and set the initial membrane potential $\boldsymbol{v}$ to 0.
17: $f_{\text{SNN}}$ = Parallel Parameter normalization($f_{\text{ANN}}$)
18: **return** $f_{\text{SNN}}$

---

approach can achieve comparable accuracies to traditional ANNs with few time steps. Notably, there is only a 1% drop in accuracy observed relative to their ANN counterparts at T=10 for ViT-S/16, T=8 for ViT-B/16, T=6 for ViT-L/16, and as early as T=4 for EVA. This trend highlights the efficiency of our conversion strategy, especially within the larger models.

Taking a closer look at the EVA model, our method achieves an impressive 88.60% accuracy at just T=4, with a negligible 1% accuracy degradation while using only 35% of the energy required by the equivalent ANN model. These results demonstrate our approach's effectiveness and suggest its potential for significant energy savings without substantially compromising accuracy, particularly in complex and larger-scale model architectures.

### 5.3 Comparison with the State-of-the-art

Our experiments on the ImageNet1k dataset have pushed the frontiers of neural network efficiency and accuracy. Table 2 provides a compelling narrative of our progress. Our method is unique in that it facilitates the conversion of Transformer models into SNNs, and it stands out for its computational frugality and high accuracy yield. This marks a significant stride over previous state-of-the-art methodologies.

Firstly, our method is designed to be more efficient than direct training approaches. Instead of starting from scratch, we leverage large pre-trained models to economize on computational efforts and achieve higher accuracy levels than traditional methods. This approach demonstrates our ability to capitalize on the intrinsic

**Table 1: Accuracy and energy consumption ratio of ECMT(Ours) on ImageNet1k dataset**

| Arch. | Accuracy/Energy | Original (ANN) | Ours (SNN) | | | | | | |
|---|---|---|---|---|---|---|---|---|---|
| | | | T=1 | T=2 | T=4 | T=6 | T=8 | T=10 | T=12 |
| ViT-S/16 | Acc. (%) | 78.04 | 0.17 | 10.66 | 62.85 | 73.22 | 76.03 | 77.07 | 77.41 |
| | Energy ratio | 1 | 0.06 | 0.15 | 0.37 | 0.59 | 0.82 | 1.03 | 1.25 |
| ViT-B/16 | Acc. (%) | 80.77 | 0.24 | 20.89 | 69.98 | 77.81 | 79.40 | 80.12 | 80.38 |
| | Energy ratio | 1 | 0.04 | 0.12 | 0.30 | 0.48 | 0.66 | 0.84 | 1.01 |
| ViT-L/16 | Acc. (%) | 84.88 | 3.62 | 75.38 | 83.20 | 84.32 | 84.60 | 84.68 | 84.71 |
| | Energy ratio | 1 | 0.04 | 0.12 | 0.27 | 0.43 | 0.58 | 0.74 | 0.89 |
| EVA | Acc. (%) | 89.62 | 2.49 | 84.08 | 88.60 | 89.23 | 89.40 | 89.45 | 89.51 |
| | Energy ratio | 1 | 0.06 | 0.15 | 0.35 | 0.55 | 0.74 | 0.93 | 1.13 |

**Table 2: Comparison between the proposed method and previous works on ImageNet1k dataset**

| Method | Type | Arch. | Param. (M) | T | Accuracy (%) |
|---|---|---|---|---|---|
| Spikingformer[54] | Direct Training | Spikingformer-4-384-400E | 66.34 | 4 | 75.85 |
| Spike-driven Transformer[48] | Direct Training | Spiking Transformer-8-768* | 66.34 | 4 | 77.07 |
| Spikeformer[29] | Direct Training | Spikeformer-7L/3×2×4 | 38.75 | 4 | 78.31 |
| RMP[18] | CNN-to-SNN | VGG-16 | 138 | 4096 | 73.09 |
| SNM[44] | CNN-to-SNN | VGG-16 | 138 | 64 | 71.50 |
| TS[9] | CNN-to-SNN | VGG-16 | 138 | 64 | 70.97 |
| QFFS[27] | CNN-to-SNN | VGG-16 | 138 | 4(8) | 72.10(74.36) |
| QCFS[3] | CNN-to-SNN | ResNet-34 | 21.8 | 64 | 72.35 |
| | | VGG-16 | 138 | 64 | 72.85 |
| SRP[20] | CNN-to-SNN | ResNet-34 | 21.8 | 4(64) | 66.71(68.61) |
| | | VGG-16 | 138 | 4(64) | 66.46(69.43) |
| MST[45] | Transformer-to-SNN | Swin-T(BN) | 28.5 | 128(512) | 77.88(78.51) |
| STA[25] | Transformer-to-SNN | ViT-B/32 | 86 | 32(256) | 78.72(82.79) |
| **ECMT(Ours)** | Transformer-to-SNN | ViT-S/16 | 22 | 8(10) | 76.03(77.07) |
| | | ViT-B/16 | 86 | 8(10) | 79.40(80.12) |
| | | ViT-L/16 | 307 | 4(8) | 83.20(84.60) |
| | | EVA | 1074 | 4(8) | 88.60(89.40) |

efficiencies of pre-trained networks and apply them successfully to SNNs.

Secondly, our technique surpasses the CNN-to-SNN conversion methods in every aspect. Remarkably, even with the ViT-S/16 model at just 8 time steps, we have achieved an accuracy of 76.0%, which outperforms the highest accuracy metrics achieved in previously published CNN-to-SNN works. This highlights the effectiveness of our conversion protocol and confirms its superiority in translating CNN architectures into their spiking counterparts.

Finally, compared to the Swin-T(BN) transformer-to-SNN conversion method mentioned in [45], our approach does not require specific transformer structures for SNN training. Instead, it enables the direct conversion of mainstream ViT models. When compared to the transformer-to-SNN conversion method in [25], our method can decrease overall energy consumption while requiring extremely

lower latency. Based on the above discussion, our process ensures quick turnaround and achieves accuracy within 10 temporal steps.

We conducted experiments using four different models, ViT-S/16, ViT-B/16, ViT-L/16, and EVA, and found that the accuracies achieved at time steps 8, 8, 4, and 4, respectively, were as follows: 76.03%, 79.4%, 83.2%, and 88.6%. The EVA model, in particular, performed exceptionally well at reduced time steps, indicating the robustness of our method and its potential to set new benchmarks in SNN performance.

## 5.4 The Effect of Multi-Threshold Neuron

To verify the effectiveness of the Multi-Threshold Neuron, we conducted an experiment to explore the model by varying the number of thresholds in the neurons. We denoted the number of thresholds as $2n$ and experimented with $n = 4$, $n = 6$, and $n = 8$. Our results, depicted in Figure 5, illustrate that as the value of $n$ increases,

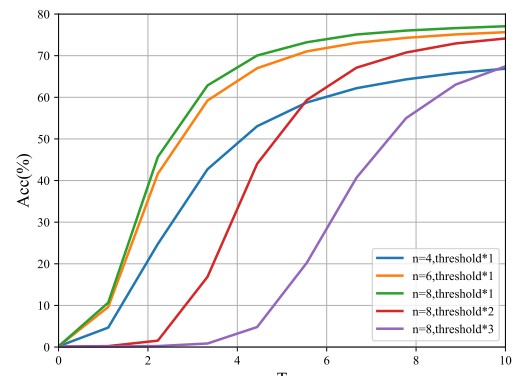

**Figure 5: Accuracy under different number and size of thresholds on ViT-S/16, $2n$ denotes the number of thresholds.**

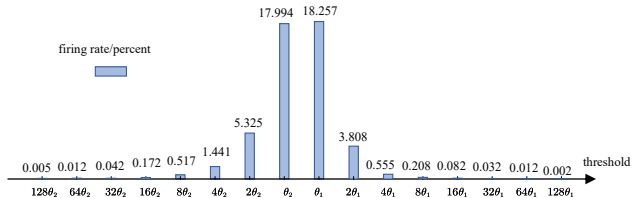

**Figure 6: Firing rate at different thresholds**

more large thresholds are included. This suggests that having large thresholds is crucial for enhancing performance.

We raised the base threshold to investigate further while keeping $n = 8$. This enabled us to examine the impact of discarding smaller thresholds. Our findings were precise: models that lacked small thresholds performed much worse than those with both large and small thresholds. Therefore, our results suggest that having a complete range of threshold sizes is crucial for achieving optimal model performance.

We also increased the base threshold to investigate further while keeping $n = 8$. This allowed us to study the effect of smaller thresholds by their omission. The results were precise: models without small thresholds performed worse than those with both large and small thresholds. Our results showed that both large and small thresholds are crucial for the model. This emphasizes the need for a larger $n$ to achieve low-latency and high-accuracy conversion.

Additionally, we measured the firing rates of spikes associated with each threshold when $n$ was set to 8. The outcomes are presented in Figure 6, which shows that the majority of spikes cluster around the base thresholds, while the spikes generated by other thresholds are minimal. This indicates that adding thresholds consumes less energy but significantly reduces the inference latency.

## 5.5 Energy Estimation

In order to determine the energy consumption of the SNNs, we begin by calculating the theoretical computational complexity for each module presented in the EVA model, as detailed in Table 3.

**Table 3: Theoretical calculation dimensions and actual numerical results of different modules, with image patches $N = 577$, channels $C = 1408$, self-attention heads $N_h = 16$, and MLP hidden layer channels $C_h = 6144$.**

| Module | Computation | |
| --- | --- | --- |
| | Complexity | Results (M) |
| LayerNorm 1 | $N * C$ | 0.81 |
| Linear $qkv$ | $N * C * 3C$ | 3431.65 |
| Matrix Product $q, k$ | $Nh * N * (C/Nh)^2$ | 71.49 |
| Softmax | $Nh * N * N$ | 5.33 |
| Matrix Product $s, v$ | $Nh * N * N * (C/Nh)$ | 468.76 |
| Linear out | $N * C * C$ | 1143.88 |
| LayerNorm 2 | $N * C$ | 0.81 |
| MLP Linear 1 | $N * C * Ch$ | 4991.48 |
| GELU | $N * Ch$ | 3.54 |
| MLP Linear 2 | $N * Ch * C$ | 4991.48 |

We then employ the formula presented in [36] to estimate the energy consumption of SNNs, as detailed in Equation (30):

$$\frac{E_{\text{SNN}}}{E_{\text{ANN}}} = \frac{MACs_{\text{SNN}} * E_{\text{MAC}} + ACs_{\text{SNN}} * E_{\text{AC}}}{MACs_{\text{ANN}} * E_{\text{MAC}}}. \tag{30}$$

Here we set $E_{\text{MAC}} = 4.6pJ$ and $E_{\text{AC}} = 0.9pJ$ according to [22].

The original network performs most of its computation in linear and matrix product layers. Our method enables us to implement linear transformations of spikes entirely using accumulations and matrix products primarily using accumulations. As a result, we can estimate the number of multiply operations ($MACs_{\text{SNN}}$) to be zero. We evaluated the total energy consumption ratio of our method compared to the original ANNs, and the results are summarized in Table 1. Our method reaches a high accuracy of 88.60% using only 4 time steps, with a marginal loss of 1% compared to the original ANNs, while consuming only 35% of the energy.

## 6 CONCLUSION AND DISCUSSION

In this paper, we propose a novel method for converting pretrained Vision Transformers to SNNs with reduced latency. This approach diverges from previous approaches focusing on converting CNNs to SNNs or directly training SNNs, our method converts pre-trained ViTs to SNNs in a low latency. It replaces various modules with a combination of Expectation Compensation Modules and Multi-Threshold Neurons, achieving significantly higher accuracy on the ImageNet dataset with very low latency compared to previous conversion methods. Moreover, the converted models exhibit substantially less energy consumption than the original ANN ViTs. Our method bridges the performance gap between SNNs and ANNs, paving the way for ultra-high-performance SNNs.

However, our current method still requires a small amount of multiplication and cannot use accumulations for implementation alone. Future work may focus on finding alternative solutions for non-linear modules to eliminate the remaining multiplications. This will make them more suitable for conversion and pave the way for further exploration of the conversion from Transformers to SNNs.

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
