# OpenReview forum: "Towards High-performance Spiking Transformers from ANN to SNN Conversion"
_acmmm.org/ACMMM/2024/Conference — MM2024 Poster_

### Official Review · Reviewer_sBuM · 2024-05-15

**Rating:** 5
**Confidence:** 2

**Summary:**

The paper titled "Towards High-performance Spiking Transformers from ANN to SNN Conversion" addresses the challenge of converting Artificial Neural Networks (ANNs) to Spiking Neural Networks (SNNs), particularly focusing on Transformers, which are inherently non-linear and thus difficult to convert. The authors propose two main innovations to achieve high accuracy, low latency, and reduced power consumption in the conversion process:
1. Expectation Compensation Module (ECM): This module uses information from previous time steps to calculate the expected output at the current time step, thereby preserving accuracy during conversion.
2. Multi-Threshold Neurons and Parallel Parameter Normalization: These techniques aim to reduce the large time steps typically required for high accuracy, thus lowering network latency and power consumption.

The experimental results demonstrate that the proposed approach achieves state-of-the-art performance with significant reductions in energy consumption, exemplified by a top-1 accuracy of 88.60% on complex datasets, with only a 1% loss in accuracy using just 4 time steps and consuming only 35% of the original power of the Transformer.

Overall Evaluation

The paper presents a significant advancement in the field of SNNs by tackling the complex problem of converting Transformer models. The proposed methods are innovative and address key limitations of previous conversion techniques. The experimental validation on a complex dataset like ImageNet1k showcases the practical applicability and effectiveness of the approach.

Questions and Suggestions
1. Scalability Tests: Have the authors considered testing the scalability of their approach with larger Transformer models or different architectures? If so, what were the outcomes?
2. Real-world Applications: Can the authors provide insights or case studies on potential real-world applications where their converted SNNs would offer significant advantages over traditional ANNs?
3.The authors may discuss the possibility of combing their architecture with direct training method and provide insights, following literatures  may serve as reference:
 1) Wu, Y., Zhao, R., Zhu, J., Chen, F., Xu, M., Li, G., ... & Shi, L. (2022). Brain-inspired global-local learning incorporated with neuromorphic computing. Nature Communications, 13(1), 65.
2) Zhao, R., Yang, Z., Zheng, H., Wu, Y., Liu, F., Wu, Z., ... & Shi, L. (2022). A framework for the general design and computation of hybrid neural networks. Nature communications, 13(1), 3427.
3) Xu, M., Liu, F., Hu, Y., Li, H., Wei, Y., Zhong, S., ... & Deng, L. (2024). Adaptive Synaptic Scaling in Spiking Networks for Continual Learning and Enhanced Robustness. IEEE Transactions on Neural Networks and Learning Systems.
4) Zhu, X., Zhao, B., Ma, D., & Tang, H. (2021). An efficient learning algorithm for direct training deep spiking neural networks. IEEE Transactions on Cognitive and Developmental Systems, 14(3), 847-856.
5)Yin, H., Zheng, H., Mao, J., Ding, S., Liu, X., Xu, M., ... & Deng, L. (2024). Understanding the Functional Roles of Modelling Components in Spiking Neural Networks. arXiv preprint arXiv:2403.16674.

**Strengths:**

Strengths

1. Innovative Solutions: The Expectation Compensation Module and Multi-Threshold Neurons introduce novel ways to handle non-linear modules and reduce latency in SNNs.
2. Energy Efficiency: The approach significantly reduces power consumption, which is a critical factor for the deployment of SNNs on neuromorphic hardware.
3. High Performance: Achieving high accuracy with minimal accuracy loss demonstrates the effectiveness of the proposed conversion methods.
4. Comprehensive Evaluation: The use of a complex and widely-recognized dataset (ImageNet1k) for evaluation provides strong evidence of the practical impact of the research.

**Limitations:**

Weaknesses

1. Complexity of Implementation: The proposed methods, while innovative, may introduce additional complexity in the implementation and training of SNNs.
2. Scalability Concerns: It is not entirely clear how well the proposed methods would scale with even larger models or different types of neural network architectures beyond Transformers.
3. Limited Benchmarking: Although the results are impressive, the paper could benefit from more extensive benchmarking against a wider range of existing SNN models and conversion methods.

**Suitability:**

2

---

### Official Review · Reviewer_1RH2 · 2024-05-22

**Rating:** 4
**Confidence:** 3

**Summary:**

To address the issue of accuracy loss when converting Transformers to Spiking Neural Networks (SNNs), an  Expectation Compensation Module has been introduced. Additionally, a  Multi-Threshold Neuron and the corresponding Parallel Parameter Normalization have been proposed to reduce network latency and power consumption.

**Strengths:**

1. The article possesses strong logical consistency and coherence.
2. The derivation process of the formulas is coherent, detailed, and clear.

**Limitations:**

1. "Since each position of the input matrix has only one effective threshold at each time, it limits the total number of input spikes, thereby restricting the total number of operations." I don’t understand this sentence, could you explain it?
2. The reviewer is not quite clear on the specific implementation of the ‘Multi-Threshold Neuron’ method.
3. "The results of experiments presented in Section 5.4 indicate that although this neuron has multiple thresholds, most of the spikes it
generated are concentrated in 𝜃1 and −𝜃2." Could you analyze the causes behind this phenomenon?
4. ”We raised the base threshold to investigate further while keeping 𝑛 = 8. This enabled us to examine the impact of discarding
smaller thresholds. Our findings were precise: models that lacked small thresholds performed much worse than those with both large
and small thresholds. Therefore, our results suggest that having a complete range of threshold sizes is crucial for achieving optimal
model performance.",  "We also increased the base threshold to investigate further while keeping 𝑛 = 8. This allowed us to study the effect of smaller thresholds by their omission. The results were precise: models without small thresholds performed worse than those with both large and small thresholds. Our results showed that both large and small thresholds are crucial for the model." The descriptions and explanations in these two sections are similar and require further clarification. Please translate this as well.

**Suitability:**

3

---

### Official Review · Reviewer_FYaZ · 2024-05-24

**Rating:** 4
**Confidence:** 3

**Summary:**

This paper introduces an innovative method for converting Transformer models from Artificial Neural Networks (ANNs) to Spiking Neural Networks (SNNs) to achieve high-performance computational and energy efficiency. It designs an Expectation Compensation Module (ECM) and Multi-Threshold Neurons (MT). These two significantly improve the converted SNN's accuracy while maintaining low latency and power consumption, comparable to the original ANN model. Experimental results on the ImageNet1k dataset show that the method achieved a top-1 accuracy of 88.60% with only 4 time steps, incurring a mere 1% loss in accuracy, and consuming only 35% of the power of the original Transformer model. This work not only demonstrates the high efficiency and low power consumption of ANN-to-SNN conversion on complex datasets but also provides a new direction for the future development of more efficient SNN models.

**Strengths:**

Efficiency: The method can achieve high accuracy within a few time steps by using ECM and MT neurons.
Low Power Consumption: Compared to traditional ANN models, the converted SNN models significantly reduce energy consumption while maintaining similar accuracy.

**Limitations:**

1. Many symbols, such as time-step T and module m, are not mathematically typeset.
2. At line 516, the authors state that one threshold can lead to excessively high firing rates or high inference latency. Please explain the specific reasons for this conclusion.
3. Why is the method in this paper tested only on ImageNet? What is the applicability on other datasets?

**Suitability:**

2

---

### Official Review · Reviewer_9C2o · 2024-05-24

**Rating:** 2
**Confidence:** 4

**Summary:**

The paper is about developing a novel method for converting pre-trained Vision Transformers into Spiking Neural Networks with reduced latency and improved energy efficiency. The authors propose a method that diverges from previous approaches that focused on converting Convolutional Neural Networksto SNNs or directly training SNNs. Instead, this method converts pre-trained ViTs into SNNs using a combination of Expectation Compensation Modules and Multi-Threshold Neurons, achieving significantly higher accuracy on the ImageNet dataset with very low latency compared to previous conversion methods. Additionally, the converted models exhibit substantially less energy consumption than the original ANN ViTs.

**Strengths:**

Novelty: The paper introduces a novel approach to converting Vision Transformers to Spiking Neural Networks, which is a relatively new and innovative area of research. The use of Expectation Compensation Modules and Multi-Threshold Neurons is a new concept that has not been extensively explored in the literature, indicating a significant contribution in terms of originality.

Clarity: The paper seems to be well-structured and clearly written, with a logical flow of ideas and concepts. The use of figures and tables to illustrate the proposed architecture and results enhances the clarity of the presentation.

**Limitations:**

1.	The Multi-Threshold Neuron section lacks an explanation for why thresholds are expanded using powers of 2 instead of other methods.
2.	Maybe the formula 19 contains errors.
3.	The formula 25 provided in the document is a piecewise function that describes the activation behavior of a Multi-Threshold Neuron. However, it appears that the specific details or context necessary for a complete understanding of the formula are missing.
4.	There is a lack of certain experiments to verify the effectiveness of the Expectation Compensation Module.
5.     There is only one dataset for validation. The performance of the converted spiking Transformer could be validated on other detection or trakcking applications.

**Suitability:**

2

---

### Official Review · Reviewer_zhhB · 2024-06-07

**Rating:** 2
**Confidence:** 3

**Summary:**

The paper introduces an approach for converting artificial neural networks, specifically Transformers, into spiking neural networks. The paper proposes the use of an expectation compensation module and multi-threshold neurons to address the challenges associated with the conversion of non-linear modules in Transformers. Experimental results demonstrate a top-1 accuracy of 88.60% on the ImageNet1k dataset with only a 1% accuracy loss and 35% of the original power consumption of the Transformer.

**Strengths:**

1.This paper is well-written and leads the reader to understand it.

2.The paper provides extensive experimental results to demonstrate the performance of this work.

**Limitations:**

1.The proposed methods, including the ECM and Multi-Threshold Neurons, introduces additional complexity in the implementation, which could be a barrier for practical deployment.

2.While the results are promising for vision tasks using the ImageNet1k dataset, it remains to be seen how well the method generalizes to other types of tasks and datasets.

3.The paper focuses on comparing the proposed method with existing SNN models but provides limited direct comparisons with other ANN-to-SNN conversion methods for Transformers, which would have strengthened the validation of the approach.

4.The paper mentions that the proposed method does not require additional training, but it does not detail the computational overhead involved in the conversion process itself, which might be significant.

5.The conversion from ANN to SNN often involves approximations that might affect the temporal dynamics of the network. The paper does not fully explore how well these dynamics are preserved and whether any significant information is lost during conversion.

**Suitability:**

1

---

### Meta-Review · Area_Chair_yLC1 · 2024-06-27

**Recommendation:** Accept (Poster)
**Confidence:** 5

**Metareview:**

All reviewers consider the method proposed in this paper to be highly significant, leading to the consideration of accepting the paper. The paper proposes an Expectation Compensation Module to improve the accuracy of ANN-SNN conversion methods. The expected output at time step 𝑇 is calculated using information from the previous 𝑇 time steps. Additionally, Multi-Threshold Neuron and the corresponding Parallel Parameter normalization are employed to overcome the issue of requiring large time steps in the conversion method. The method not only achieves high accuracy on complex datasets with only 1% accuracy loss in ANN-SNN conversion, but also the converted model has the advantage of low power consumption.